# Effect of Surface Modification via Laser Irradiation on the Operability of Carbide End Mills When Cutting Aircraft Alloys

**Andrey V. Gusarov** , **Mars S. Migranov \*** , **Artem P. Mitrofanov, Andrey S. Gusev** , **Artur M. Migranov** and **Roman S. Khmyrov**

Department of High-Efficiency Processing Technology, Institute of Production Technology and Engineering, Moscow State University of Technology "STANKIN", Vadkovskiy per. 3A, 127055 Moscow, Russia
* Correspondence: migmars@mail.ru; Tel.: +79-613-642-534

**Abstract:** In modern aviation production, innovative hard-to-machine materials with unique physical and mechanical properties are being used increasingly. When processing such materials, the weakest link in the technological chain of production is the metal-cutting tool. In this paper, to improve the efficiency of the blade cutting of heat-resistant alloys, we propose the use of nanostructured multilayer wear-resistant coatings with subsequent laser processing of the cutting surfaces of the end milling cutters according to various schemes. In this case, an increase in the wear-resistant properties of the cutting edge by 15%–20% is provided due to the formation, at high temperatures, of secondary structures with increased wear resistance and a reduction of the temperature and force loading of contact processes. Methodologically, the work was carried out in several consecutive stages: the first stage was the determination of effective grades of wear-resistant coatings obtained via various installations with their subsequent laser processing during the «SharpMark™ Fiber» installation; at the second stage tribotechnical tests were carried out during the tribometer and adhesion installation; and in the third stage wear-resistant, temperature-force tests were carried out using milling machines in various cutting modes. According to the results of the field tests, the tool durability period was increased by 15%–20%.

**Keywords:** carbide cutter; laser machining; heat-hardness and heat-resistant chromium-nickel alloy; tribotechnical tests; wear resistance of cutting tools; temperature and components of cutting force

## 1. Introduction

For the further development of machine-building production in contemporary conditions, the use of innovative processes for the blade-cutting machining of high-strength and heat-resistant machined materials used in parts of assemblies (working under conditions of increased temperature, force loads, and aggressive media) requires a toughening of the technological modes of operation of metal-cutting tools. In such conditions, the machining tool should ensure the fulfillment of strict requirements, in particular: increased wear resistance in all processing modes; a high strength under high pressure and increased temperatures of the cutting edge; and an economic and technological design. It is known [1–3] that the issues regarding increasing the efficiency of cutting machining in the presence of modern high-performance metal-cutting equipment, equipped with expensive systems of mechatronic, adaptive, and numerical program control, with the mandatory provision of high-quality indicators of the machined surface, to a large extent depends on the operational properties of metalworking tools, such as the period of durability, heat and crack resistance, resistance to shock and vibration loading. At the same time, the operability of the metal-cutting tool and its efficiency in the conditions of modern machine-building production is a real and important problem. Currently, there are various trends and approaches to this, but the main and determining parameters of metalworking tool efficiency are still a complex of its physical and mechanical properties and the provision of rational or optimal machining modes with minimization of wear intensity.

According to numerous studies [4,5], to increase the efficiency and technological reliability of cutting machining, it is necessary to increase the difference between the tool durability period and its machining time, for example, by reducing the requirements for machining accuracy, reducing the machining modes, increasing the cost of tool materials (by replacing the tool after one pass or operation), reducing the intensity of tool wear (by using progressive coolants and multifunctional wear-resistant coatings), or active control by physical parameters (temperature stabilization at an optimum level). In this case, the last two are the most acceptable, since the reduction of cutting modes leads to an increase in the durability period of the monotonic dependence of durability on speed alone. Hard-to-machine materials are characterized by a non-monotonic relationship between cutting speed and durability [6–8], and a reduction in speed below a certain level is not accompanied by an increase in durability. An increase in the number of tool changes is limited in semi-finishing and finishing operations due to the requirement of high quality of the surface and subsurface layers, because of which it is impossible to change the tool before the end of the pass. A reduction in accuracy requirements, which is expressed in the increase of tolerances on the dimensions of manufactured parts, is possible, if we assume a selective assembly of machine units, as is the case in the production of rolling bearings.

At present, there are various ways to increase the wear resistance of cutting tools, and the most effective and least costly is the use of innovative nanostructured multilayer composite coatings with a subsequent combined surface treatment using various electrophysical effects. Based on the above, this work aims to study the effect of surface modification via laser irradiation on the serviceability of carbide end mills when cutting aircraft alloys.

Based on the above, this work aims to develop innovative nanostructured multilayer coatings on carbide end mills for machining hard-to-machine aircraft alloys and to study their ex-operational properties, as well as to improve further the wear resistance of cutting tools through modifying their surface via laser treatment.

## 2. Theoretical and Experimental Prerequisites

When cutting metals, the spatial shape of the contacting surfaces changes due to the wear of the tool material. This, even in the case of blade machining via cutting, with constant elements of modes using lubricating and cooling technological methods and modern wear-resistant coatings, leads to an increase in cutting temperature and stresses on the actual contact zone of the cutting edge, which, in general, contributes to a significant reduction in its wear resistance. Thus, contact wear, the distribution of specific loads and temperatures, is a mutually conditioned process, which should be taken into account when developing methods of intensification of the blade-cutting process, in particular when developing and applying wear-resistant coatings. According to numerous studies [2,9–11], the surface properties of the coated cutting tools should exceed the properties of the base material. At the same time, the coating composition is selected in such a way that the following requirements are met [10]: the ability to resist fracture and retain its properties under conditions of high pressure and high temperatures; the similarity of temperature requirements for characteristics such as the linear expansion coefficient, Poisson's ratio, the elastic modulus, etc., for the coating and the cutting tool; and the coating material should reduce the tendency of the tool material to adhere to the material of the workpiece, but at the same time, the promote the adhesive bonding of the coating material to the material of the workpiece.

The main directions of improvement of coated cutting tools are determined by a complex of factors, which are related to the advancement of properties of coatings and composition «coating—tool base», as well as to the optimization of conditions of use of coated cutting tools and the possibility of further modification of wear-resistant coatings by exposure (treatment) by various electro-physical methods (laser treatment, induction current treatment, surface hardening by high-frequency currents, etc.). Combined hardening treatment is an effective method of increasing the serviceability of coated cutting tools. Increasing the durability of the coating and performance of the cutting tool can be achieved

by introducing an intermediate layer between the coating and the tool substrate to increase the resistance of the matrix to thermoplastic deformations and reduce the stress gradient at the boundary between the coating and the tool substrate [12]. The versatility of ion nitriding in glow discharge plasma and the principal possibility of combining the nitriding and coating processes in one unit make these methods the most suitable for creating an intermediate layer of increased hardness and heat resistance. To improve the serviceability of cutting tools, it is suggested to treat the surface of cutting tools with powerful current pulses [2] before coating, to subject cutting tools to nitroxidation [6], which helps to increase the stability of the cutting edge to deformations at high temperatures and loads during cutting. Laser pretreatment of the substrate with subsequent coating increases the hardness and heat resistance of the tool base, which reduces the tendency of the cutting edge to elastic-plastic deformations and increases the durability period of the cutting tool by 1.5–2 times [13]. Despite the high performance of cutting tools as a result of the application of the above methods of combined hardening treatment, these methods improve the properties of the tool base only and do not change the properties of the coating, which does not provide a sufficiently high efficiency of cutting tools and in most cases is acceptable only for steel tool material.

In this case, the authors justify the use of individual elements as wear-resistant coatings, as well as the possibility of applying various additional effects on the cutting edge only at the qualitative level: choosing them based on chemical affinity with the outer layer of the coating and not taking into account the complex phenomena of the structural-phase state of the composition at high temperatures, adhesion and diffusion in the contact zone.

It is known that multilayer coatings must meet the following requirements [2,11–13]: the layers directly adjacent to the tool base must provide bonding strength; the upper layers must have increased hardness and strength, as well as minimum compatibility with the specific material being processed. Using such a coating, in which as a top layer are used materials that have a minimum activity with the material to be processed, allows for a significant increase in the durability period of the cutting tool. Thus, the increase in serviceability of coated cutting tools can be provided by increasing the microhardness, strength, and crack resistance of coatings. At the same time, it should be noted that high microhardness, strength, and crack resistance of layers cannot fully ensure high productivity of cutting tools. Having high values of these characteristics, the layer may have low adhesion strength to the tool base, which may eventually lead to flaking of the layer during cutting and, ultimately, to insufficient performance of the cutting tool. Therefore, one of the conditions for high wear resistance of the coating on the working surfaces of the cutting tool is the high strength of its adhesion with the substrate. As noted above, it is possible to increase the adhesion strength of the coating and the tool substrate by applying additional processing of the cutting tool with the layer, such as laser processing, etc.

In the noted works, there is practically no data substantiating the practicality and efficiency of the application of additional laser surface treatment of innovative multilayer nanostructured wear-resistant coatings on carbide cutting tools to increase their wear resistance due to reduction of temperature and force loading and improvement of contact processes during processing of heat-resistant and heat-hardness alloys based on chromium and nickel.

The article, during the development and research of wear-resistant coatings with post-laser treatment (modification) of the surface of cutting tools for blade processing of chromium-nickel alloys, was supposed to increase the operational properties of carbide milling cutters due to:

— obtaining nanostructured multilayer coatings with an additional layer of titanium diboride on carbide cutters;
— formation of such a texture on the surface of the cutter with wear-resistant coating by laser treatment, which would allow, firstly, to improve adhesion between the coating and the substrate of the tool material; secondly, to reduce the temperature and force loading of the contact interaction; thirdly, to promote the formation of strong

amorphous-like solid and lubricating structures on the contact surfaces under specific temperature and force conditions.

## 3. Materials, Equipment, and Technology of Experimental Studies

Experimental studies were carried out at the milling of heat-resistant and heat-hardness chromium-nickel alloys EP99 and EK61, used in critical parts of aviation equipment units with the use of carbide four-peripheral end mills with diameter (12 and 20 mm) grade H10F, the chemical composition of the materials under study are given in Table 1.

**Table 1.** Chemical composition of machined and tooling materials.

| Chemical Element | Percentage of Content, (wt%) | Chemical Element | Percentage of Content, (wt%) |
|---|---|---|---|
| EP99 | | | |
| C | $\leq 0.1$ | Al | 2.5–3.2 |
| S | $\leq 0.015$ | V | $\leq 0.005$ |
| P | $\leq 0.015$ | Ti | 1–1.5 |
| Mn | $\leq 0.4$ | Mo | 3.5–5 |
| Cr | 18–20 | Nb | $\leq 1.5$ |
| Si | $\leq 0.5$ | Ce | $\leq 0.02$ |
| Ni | Remnant | W | 6–7.5 |
| Fe | $\leq 5$ | Co | 5–8 |
| Cu | 0.07 | – | – |
| EK61 | | | |
| C | $\leq 0.05$ | Al | 0.5–1.3 |
| S | $\leq 0.015$ | V | 0.3–0.6 |
| P | $\leq 0.015$ | Ti | 0.2–1 |
| Mn | $\leq 0.7$ | Mo | 3–5 |
| Cr | 14.5–18.5 | Nb | 4–5.5 |
| Si | $\leq 0.5$ | Ce | 0.02 |
| Ni | Remnant | Zr | $\leq 0.1$ |
| Fe | 12–16 | La | 0.01 |
| Cu | 0.1–1 | – | – |
| H10F | | | |
| Co | 10 | – | – |
| WC | 90 | – | – |

The application of innovative multicomposite multilayer coatings with additional surface modification of metal-cutting tools by laser treatment was used, both modern units of «Platit π311» and «Platit π411» series and modernized unit «NNV-6.6-I1» additionally equipped with magnetic-arc filtration and temperature control systems, a unit with magnetron system used in machine-building production. According to the chemical composition of the coating: (TiCrAl)N; (TiAlSi)N; (ZrCrAl)N; (ZrMoAl)N; (ZrMoHfCrAl)N; $TiB_2$; nACo3; nACRo; nACo3+epilama; nACo3+$TiB_2$; nACRo+$TiB_2$; nACo3+$TiB_2$+epilama; (CrAlSi)N + epilama; $TiB_2$+epilama; nACRo+epilama; nACRo+$TiB_2$+epilama; (CrAlSi)N; (CrAlSi)N with DLC (Diamond-Like Carbon); (CrAlSi)N with DLC + epilama.

In this work, to further increase the wear resistance of end mills with wear-resistant coatings, the cutting edges of the milling cutter were treated with a multifunctional laser complex «SharpMark™ Fiber» (Figure 1) at different power levels and hardening treatment modes before coating (without coating) and after the application of wear-resistant coatings (Table 2).

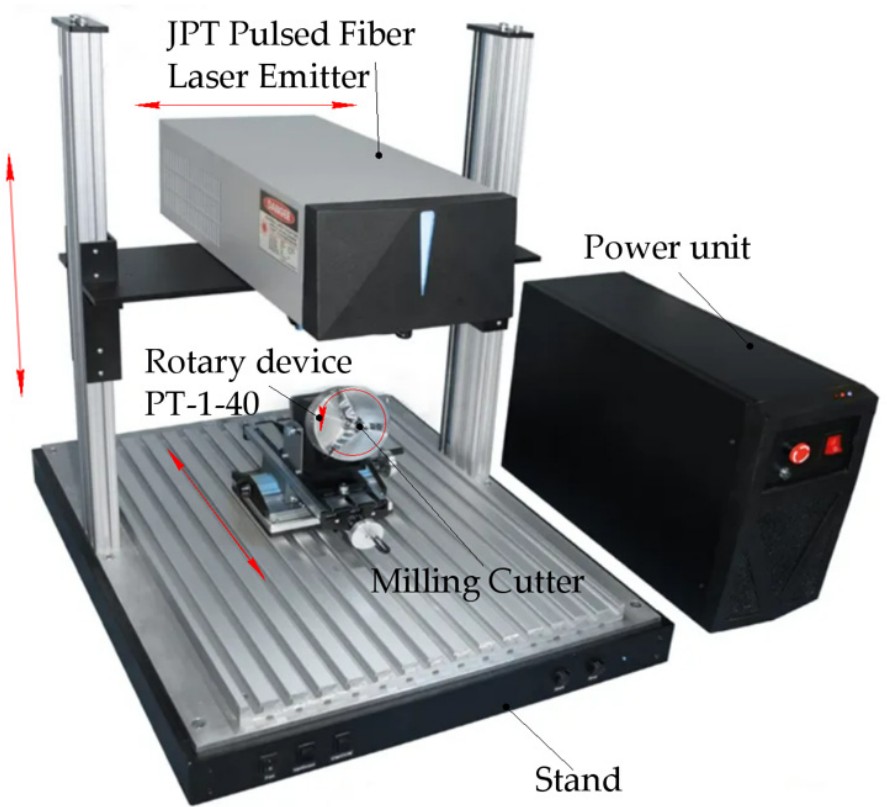

**Figure 1.** Multifunctional SharpMark™ Fiber laser system.

**Table 2.** Laser treatment modes for indenter and cutting tool surfaces.

| Power, W | 10; 20; 30; 50 |
|---|---|
| Spot diameter, μm | 25; 40; 80 |
| Pulse duration, ns | 120; 125; 150; 200 |
| Wavelength, nm | 1064 |
| Energy in the radiation pulse | up to 1 mJ |

Experimental studies were carried out at milling with carbide-alloyed four-peripheral end milling cutters of diameter (12 and 20 mm) of H10F grade without coating, with different layers and coatings after laser processing according to varying schemes in Figure 2 and various parameters of laser processing (when processing along the cutting edge: step between laser lines—b = 100, 125, 150 μm, at the width of the laser processing trace h = 30, 40, 50 μm; when processing perpendicular to the cutting edge: step between laser lines—v = 50, 70, 90 μm, at the width of the laser processing trace w = 20, 30 μm).

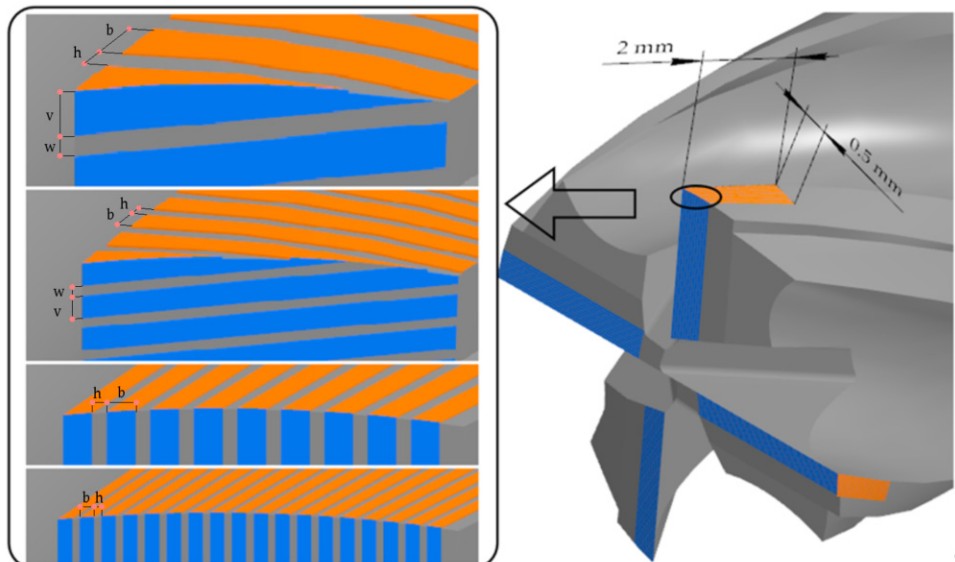

**Figure 2.** Laser processing schemes h = 20–50 μm, b = 80–200 μm («orange»—the back surface of the cutting edge; «blue»—the back surface of the end cutting edge.

Laser treatment was applied to indenters for tribotechnical tests on the Nanovea TRB tribometer and the adhesion unit (Figure 3), as well as to end mills according to different schemes.

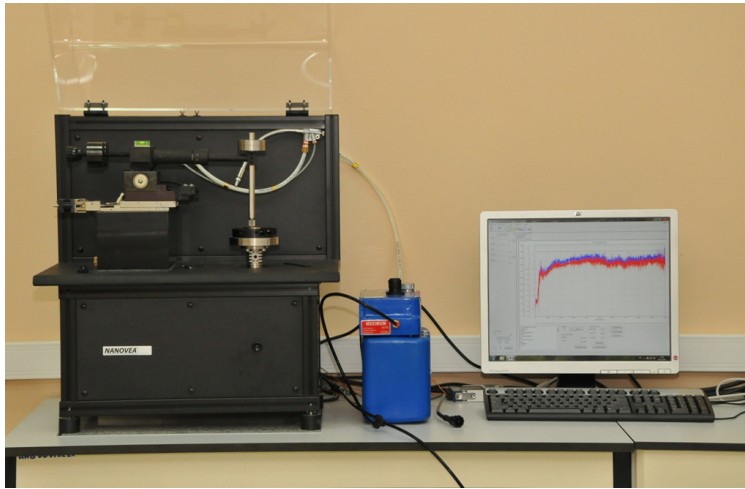

**Figure 3.** Nanovea TRB high-temperature tribometer.

At the preliminary stage, to reduce the number of full-scale experimental studies on milling, as well as to optimize the technology and modes of laser processing, a series of tests were conducted on two functionally different types of tribometer: to assess the adhesion of the coating with the substrate of the tool material; to assess the application technology and the influence of «coating,» «coating + laser treatment» and simply «laser treatment» on the tribotechnical characteristics were conducted a series of experimental studies. Tribotechnical characteristics of coatings and technologies of their application were evaluated on two types of equipment: high-temperature tribometer «Nanovea» TRB (Figure 3) and adhesion unit [13] (adhesionomer) with a circuit diagram (Figure 4).

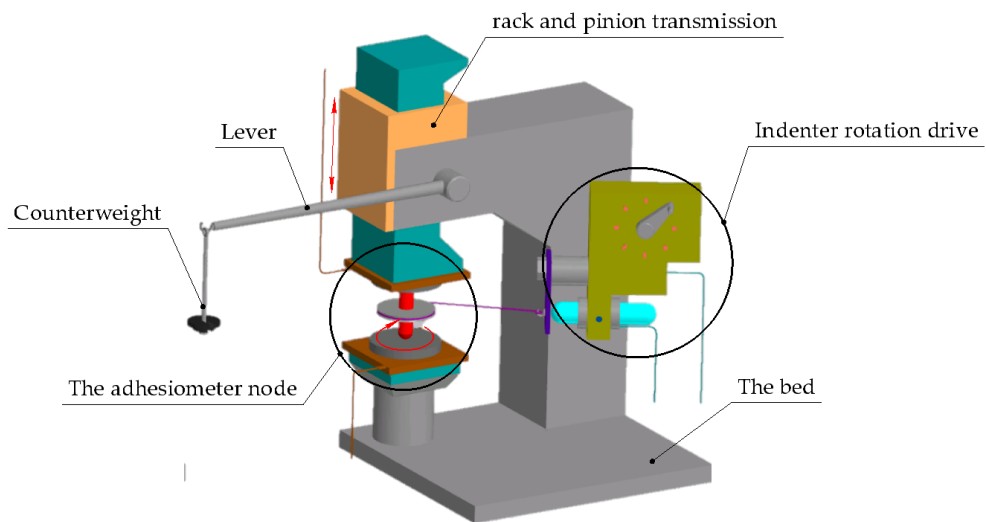

**Figure 4.** Circuit diagram of the adhesion unit.

Tribometer «Nanovea» TRB allows the evaluation of tribological properties of surfaces and coatings in a wide temperature range in different environments and lubrication modes in following DIN 50324, ASTM G99, and G133 standards. Instrument control, control of the test process, and analysis of the obtained results are performed in the specialized software TRIBOX. The counter body is a hemisphere (6 mm in diameter) made of tool material H10F with various coatings and laser treatment, and the schematic diagram of the adhesive interaction is shown in Figure 5.

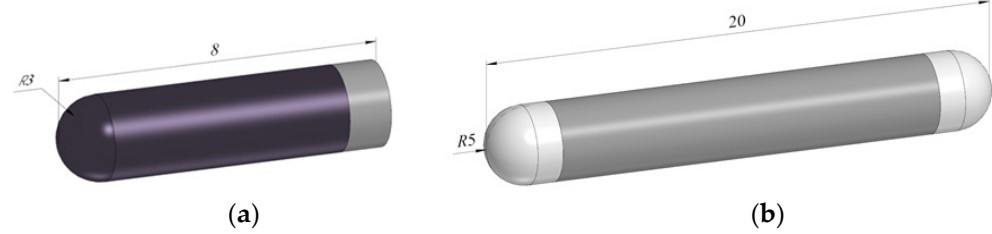

**Figure 5.** Counter body (**a**) and indenter (**b**) with coatings and laser texture for testing on tribometer and adhesionomer.

Samples of the material to be processed and spherical indenters with various laser-treated tool material coatings, as well as the sample and indenter mounting scheme, and the adhesion setup, are shown in Figure 6.

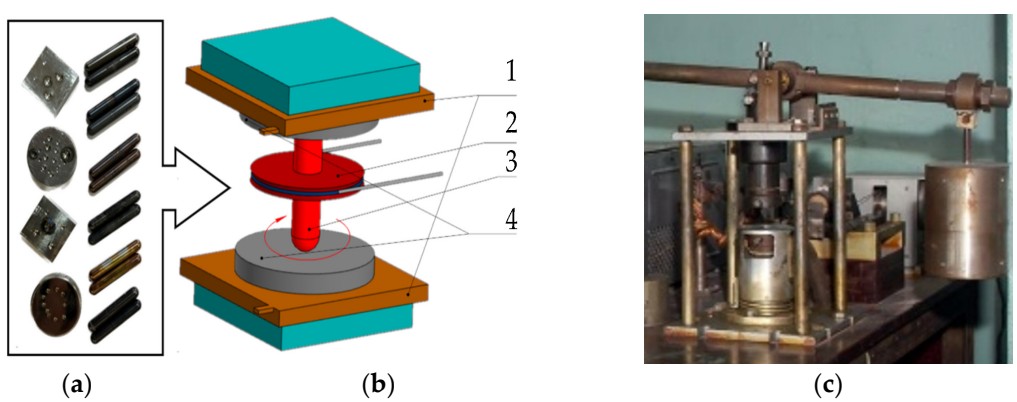

**Figure 6.** Samples of the material to be treated and spherical indenters with different coatings (**a**); schematic (**b**) of the setup and adhesionomer (**c**).

To reduce the time to determine the effective composition of coatings and modes of their application, as well as the cost of both tool and cathode material in production conditions, the most informative, practical, and with minimal time consumption is to conduct a series of adhesion tests [14]. The estimation of tribotechnical parameters ($\tau_{nn}$, $p_{rn}$, $\tau_{nn}/p_{rn}$) in the experimental method [15] was used. This method is based on a physical model, which, in the first approximation, reflects the actual conditions of friction and wear at the local contact. According to this model, a spherical indenter (Figure 6c) made of tool material with different coatings (simulating a single irregularity of the contact spot of rubbing solids), squeezed by two plane-parallel samples made of machinable material (with high accuracy and cleanliness of the contacting surfaces) rotates under load around its axis. The force $F_{exp}$, spent on the rotation of the indenter and applied to the cable placed in the groove of the disk, is mainly related to the shear strength $\tau_{nn}$ of the adhesive bonds.

Based on the above, we have developed a setup [2,15] (Figure 7), which, in the first approximation, reflects the actual conditions of friction and wear at local contact. According to this model, a spherical indenter (1) made of tool material H10F with various coatings (simulating a single non-uniformity of the contact spot of rubbing solids), clamped with two plane-parallel samples (10) made of chromium-nickel alloys—EP99; EC61 (with high accuracy and cleanliness of the contacting surfaces), rotates under the load N obtained by a lever mechanism (13) with weights (12) around its axis. The force $F_{exp}$ expended on the rotation of the indenter (1) mounted on the disk (2) by applying to the cable (4) is mainly related to the shear strength τnn of the adhesive joints. The other end of the cable is connected to the elastic element (6) and, through the sensor (9), is recorded on the recording device (8) and is driven through the skid (7) from the rotation drive (5). To ensure the temperature regime in the contact zone of the indenter (1) with samples of the processed material (10) in a wide range of variation (0–1300 °C) is used electro-contact method from power-controlled transformers (14,15) at the same time high voltage is supplied to the terminals (11) isolated (3) from the body. For temperature control by regulating the current and voltage, ammeter (16) and voltmeter (17) are used, respectively.

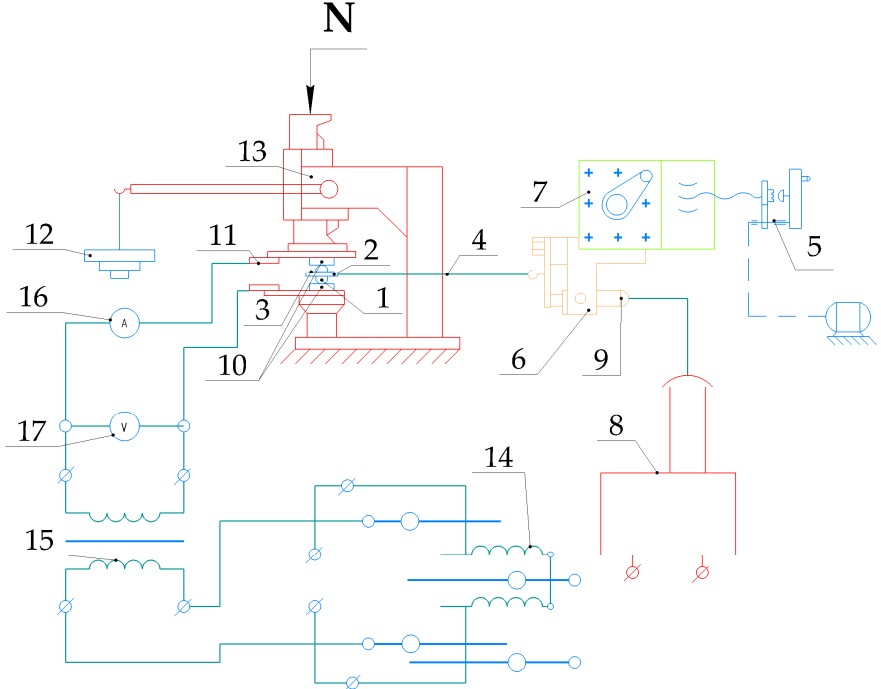

**Figure 7.** Principle scheme of the installation for adhesion research. 1—indenter; 2—rotating disk; 3—insulator; 4—cable for rotation; 5—rotation drive; 6—elastic element; 7—skid; 8—recorder; 9—sensor; 10—samples; 11—copper plates with current collector; 12—set of weights; 13—mechanical press; 14—current regulator; 15—transformer; 16—ammeter; 17—voltmeter.

The shear strength $\tau_n$ of adhesive bonds is determined from the ratio

$$\tau_n = \frac{3}{4} \cdot \frac{F_{exp}}{\pi} \cdot \frac{R_{exp}}{r_{pr}^3}, \tag{1}$$

where $F_{exp}$—the circumferential force on the disk rotating the indenter;
$R_{exp}$—radius of the disk in which the indenter is fixed;
$r_{pr}$—radius of the indentation (well) on the samples.

Due to the small size of the indentation, it is assumed that the normal stresses acting on the sphere's surface are constant and equal in the region of the entire indentation. They are calculated as follows:

$$p_r = \frac{N}{n \cdot r_{pr}^2}, \tag{2}$$

The adhesion (molecular) component of the coefficient of friction is equal to:

$$f_M = \frac{\tau_n}{p_r} = \frac{3}{4} \cdot \frac{F_{exp}}{N} \cdot \frac{R_{exp}}{r_{pr}} \tag{3}$$

Particular attention should be paid to the roughness, cleanliness, and accuracy of the surfaces of the contacting bodies. The load N is selected so that in the zone of contact, the average stresses at the connection are equal to the Brinell hardness of the material of flat samples. In this case, as shown in the works [14,15], at relative embedding, flat plastic contact and some «spreading» of surfaces are provided, leading to the breakage of foreign coatings (oxide and sorbed films) and the connection of pure (juvenile) metal surfaces.

$$0.02 \leq \frac{h}{r_1} \leq 0.2 \tag{4}$$

Geometric dimensions of samples ($\varnothing$ 20 mm, thickness—7 mm), indenter ($r_{sf} = 2.5$ mm, height—25 mm), and dwell time (about 1 min) under average load are determined by technical requirements for similar samples during Brinell hardness testing (GOST 9012-59). To ensure uniformity of physical and mechanical properties, the selections are made from one batch of material.

Experiments conducted under conditions of plastic contacting (but at $h/r_1 \leq 0.2$) showed [14,15] that the tangential strength of the adhesive bond and the adhesion component of the friction coefficient does not depend on either the load or the indenter diameter.

With the method described above, it is possible, in principle, to determine the dependence of $\tau_n$ on normal stresses either by using the «artificial substrates» method or by introducing the indenter during elastic contacting.

Thus, this estimation method of $\tau_n$ и $f_M$, in our opinion, most fully meets the requirements, which should be completed by the process of laboratory measurement of adhesion indices (at the spherical interpretation of irregularities of rubbing surfaces).

After preliminary tests on tribometers, a series of experimental studies of wear resistance of cutting tools with wear-resistant coatings and subsequent laser treatment, as well as temperature and force conditions during milling according to the principal scheme (Figure 8), were carried out.

In full-scale tests, according to numerous recommendations, as well as the modes used at the enterprises, testing for wear resistance of cutting tools and machinability of heat-resistant chromium-nickel alloys were carried out at spindle speed (n = 800 rpm), minute tool feed (S = 65 mm/min), width of the cut layer ($a_e$ = 4 mm) and depth of the cut layer ($a_p$ = 1 mm). To observe the identity of the obtained results of wear-bone tests and to ensure the accuracy of measuring the value of the wear chamfer of milling cutter teeth on the rear surface ($h_r$, mm), two types were used simultaneously: optical-mechanical microscope «MIR-2M» with a reference system MOB-I5 (with a measurement accuracy up to 0.002 mm) and mounted on the machine bed; for photo fixation of the wear chamfer

and additional control when the wear value reaches 0.3–0.35 mm—universal electronic stereo microscope «Carl Zeiss Stereo Discovery V12» with telecommunication system and providing visualization of the object with the camera «Zeiss Axiocam 503 Color» 3 mp. To obtain a complete picture (dependence) of tool wear on the length of the cutting path, as well as the need to take into account all stages (sections) of the wear curve (pre-work, typical, and critical), the measurement was carried out at a partial (equal) number of passes of milling cutters.

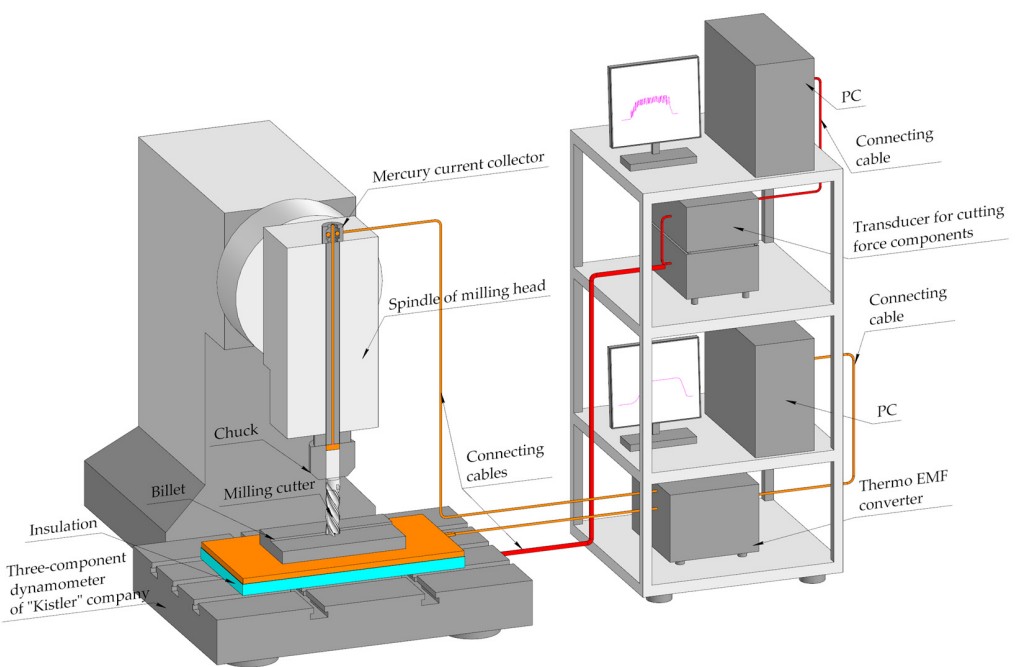

**Figure 8.** Principal scheme of measuring temperature and components of cutting force on milling machine VM-127M.

For comparative analysis of the obtained results of wear-resistance tests, the following were plotted: dependences of the effect of the cutting path length ($l$, m) on the wear on the rear surface ($h_r$, mm); dependences of the durability period ($T$, min) on the applied wear-resistant coating at the corresponding blunting criterion—$h_r^{cr} = 0.3$ mm, as well as dependencies of the critical cutting path length ($l_{cr}$, m) on the applied wear-resistant coating. At the same time, the durability period ($T$, min) was determined as the ratio of the critical length of the cutting path ($l_{cr}$, m) to the cutting speed ($V$, m/min) or as follows:

$$T = \frac{l_{cr}}{V} \tag{5}$$

During milling, high cutting speeds and high temperatures in the cutting zone affect tool durability, and it depends on several reasons: technological—cutting modes; material science—various properties of the machined material (hardness, toughness, thermal conductivity, and ability to unstrengthen when heated). During operation, cutting tools are in a complexly stressed state. First, they experience very high contact stresses and pressures on the working edge, which are necessary to deform or fracture (cut) the material to be machined. In this case, the working edge is in conditions close to non-uniform all-round compression, which puts the metal in a more plastic state due to the increasing proportion of tangential stresses. At very high tensions, especially if they are accompanied by heating, deformation and plastic flow of the thin surface layer can be observed [16–19]. So for a cutting tool without contact plastic deformation to shear the required volumes of the machined material, the hardness of the tool material must significantly (more than 2.5 times) exceed the hardness of the machined material. Tools are also subjected to in-

creased stresses, most often bending and torsion; the maximum bending or torsional torque occurs mainly in areas somewhat distant from the contacting surface, e.g., at the base of the tooth of a milling cutter, tap, etc. The ability of cutting tools to resist various stresses (bending, torsion, tension), as well as dynamic loads without brittle fracture and plastic deformation, is determined by their strength and impact toughness. Therefore, it must have sufficient mechanical strength in bending, tensile, and torsion, and impact toughness [17,20–24]. The tool can work under alternating (cyclic) loads (milling, intermittent cutting, continuous cutting with variable allowance or solid inclusions on the workpiece surface, etc.). A desirable requirement for the tool material, along with mechanical strength in compression and bending, is high resistance to fracture under alternating loading (high endurance limit). During the cutting process, the tool contact areas are exposed to high-temperature effects (up to 800–1200 °C), which can lead to thermal de-hardening and loss of hardness of the tool material. Therefore, the following essential requirement for the tool material is its ability to retain its hardness and strength characteristics at elevated temperatures corresponding to cutting temperatures. Usually, this property of the tool material is called heat resistance, which is the most important quality indicator of the tool material. Considering the need to use the tool in conditions of periodic temperature changes (for example, intermittent cutting), the tool material should be insensitive to cyclic temperature changes. The above properties determine the wear resistance of the cutting tool, i.e., the ability to resist the removal of its particles during contact interaction. Along with the requirements for the physical-mechanical and thermophysical properties of the tool material, a necessary condition for achieving sufficiently high cutting properties of the tool is a low physical and chemical activity of the tool material to the processed material, which is achieved by applying coatings [25–28].

Many works have been devoted to the study of cutting temperatures during blade machining and, various methods of their determination, from the simplest (pyrometers) to ultramodern (thermosetting), have been developed. The method of semi-artificial micro thermocouples allows to study of the temperature state of a cutting tool with sufficient accuracy and quality. Still, controlling the form of an accurate cutting tool is unsuitable for technical reasons. The use of thermal imaging cameras in milling conditions is not possible due to the non-stationarity of the instrument in its position, the concealment of the most heated zones of the mechanism by the processed material, at the high cost of these devices. To measure the cutting tool temperature, the methods of natural and artificial or semi-artificial thermocouples are most often used. The natural thermocouple is used for - the evaluation of the machinability of metals by cutting, the so-called physical optimization of modes, study of the influence of cutting process parameters on the thermal stress of cutting tools. It is assumed that a natural thermocouple measures the average temperature at the contact areas of the cutting tool.

A considerable number of works [28,29] have been devoted to studies of temperature and methods of its measurement during metal cutting, in which it is shown that to obtain information about the average contact temperature during cutting with sufficiently high accuracy, simplicity, and reliability, it is possible to use the method of a natural thermocouple. In the research, this method was used to determine the average cutting temperature. In this case, the workpiece and the cutter were isolated from each other to exclude errors from the so-called «parasitic» thermo-EMF. The thermal EMF was measured in the interval of 10–15 s after the beginning of milling. A mercury current collector, a digital voltmeter «Elemer», and a PC were used to record and record the value of thermal EMF. The schematic diagram of temperature studies during milling is shown in Figure 8.

For force experiments, a three-component dynamometer of the «Kistler» company was used–in the form of a bed for fixing the workpiece from the material being processed. The software «DynoWare» was used for data collection and analysis. «Kistler» «DynoWare» is universal and easy to use and is compatible with dynamometers or single and multi-component force sensors. When analyzing the signal, «DynoWare» provides continuous visualization of the measured curves and has all necessary mathematical and graphical

functions. In addition to the simple configuration of the most important measuring instruments, DynoWare supports the documentation of measurement processes and the storage of configuration and measurement data.

## 4. Experimental Results and Their Discussion

For tribotechnical tests on the Nanovea TRB tribometer and high-temperature adhesion unit, indenters were manufactured, and innovative wear-resistant coatings were applied with subsequent laser treatment (texturing), shown in Figure 9. The specimens were studied using a Tescan Vega 3 LMH scanning electron microscope (SEM, Tescan, Brno, Czech Republic) equipped with an X-Act energy dispersive analysis (EDX) module (Oxford Instruments, Abingdon, UK). Images were obtained at the accelerating voltage of 10 kV using a secondary electron detector (SE) and a backscattered electron detector (BSE). For this purpose, an archetype of laser processing (texturing) was developed in the form of longitudinal parallel lines (a,b) and concentric circular lines (c) with optimization of their pitch and width of the laser beam (Figure 10).

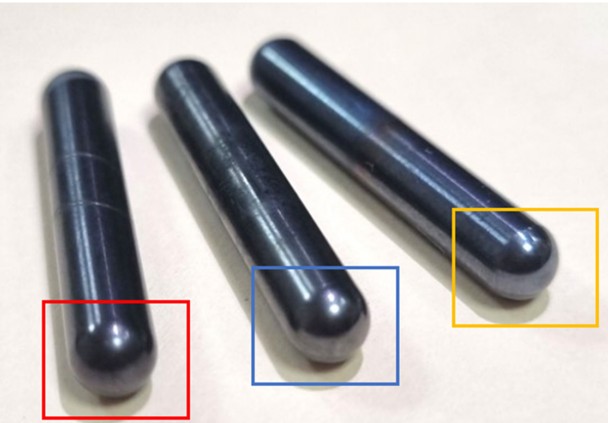

**Figure 9.** Samples of indenters with wear-resistant coatings and after laser processing (texturing).

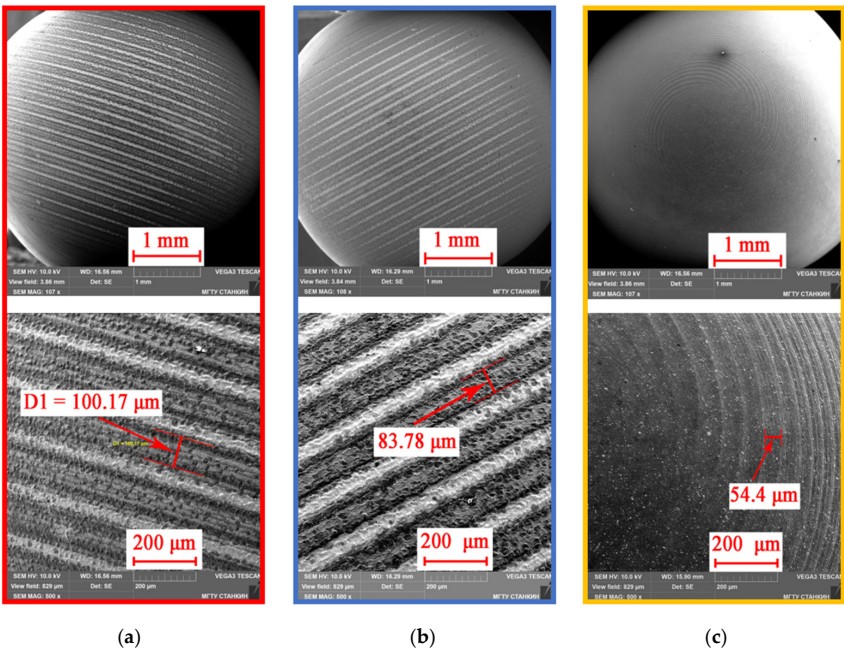

(**a**)  (**b**)  (**c**)

**Figure 10.** Laser-textured counter body (indenter) for Nanovea TRB tribometer with wear-resistant coating after laser-treatment coated and laser-textured indenters for adhesion testing. Longitudinal parallel lines (**a**,**b**) and concentric circular lines (**c**).

The cutting part of milling cutters with wear-resistant coatings was laser-treated with different texturing parameters for subsequent experimental wear-resistance tests and temperature-force dependence studies (Figures 11 and 12).

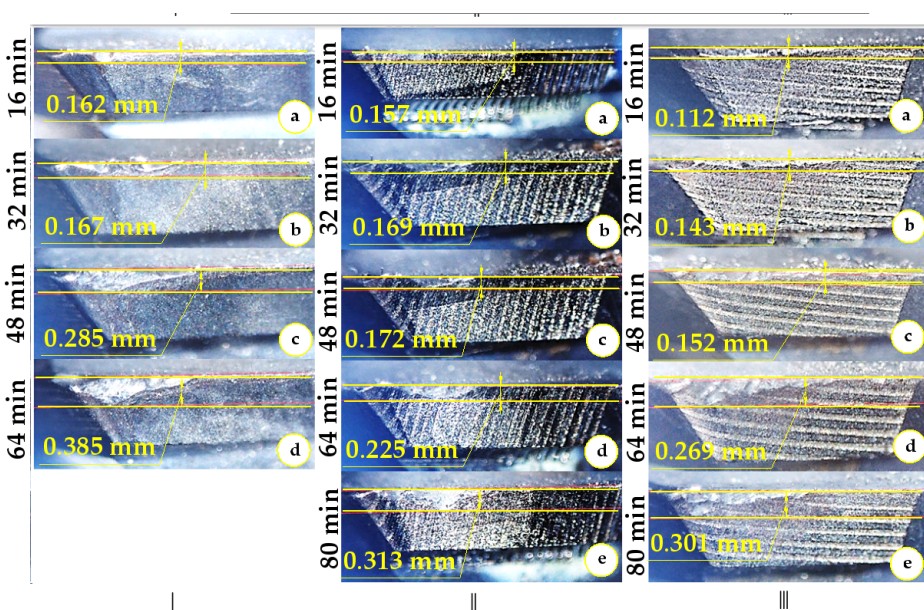

**Figure 11.** I-untextured, II-perpendicular, III-parallel. Wear kinetics on the rear surface: (**a**) longitudinal texture on the end top of the cutter ($p$ = 10 W; b = 0.75 mm); (**b**) back surface longitudinal texture of the cutting edge ($p$ = 20 W; b = 0.3 mm); (**c**) back surface longitudinal texture of the cutting edge ($p$ = 20 W; b = 0.285 mm); (**d**) longitudinal texture of the back surface of the end cutting edge ($p$ = 20 W; b = 0.3 mm); (**e**) back surface longitudinal texture of the cutting edge ($p$ = 20 W; b = 0.633 mm).

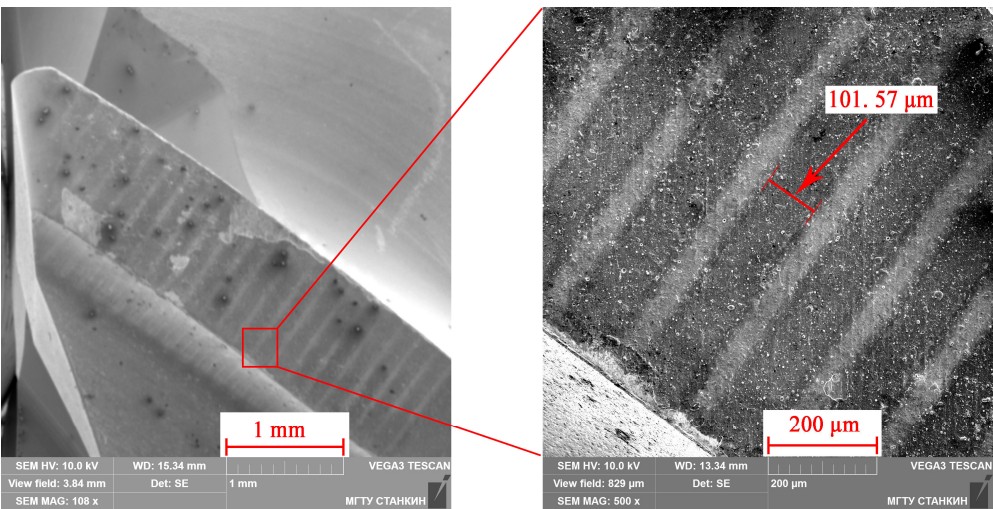

**Figure 12.** The transverse texture of different cutting-edge surfaces with wear element.

The results of tribotechnical studies on the high-temperature tribometer «Nanovea TRB» are presented in Figures 13 and 14. According to the effects of the analysis of the influence of laser processing power on friction coefficient, it is established that this dependence has an extreme character for both pairs «EP99-H10F» and «EK61-H10F» with a minimum at power 20 W.

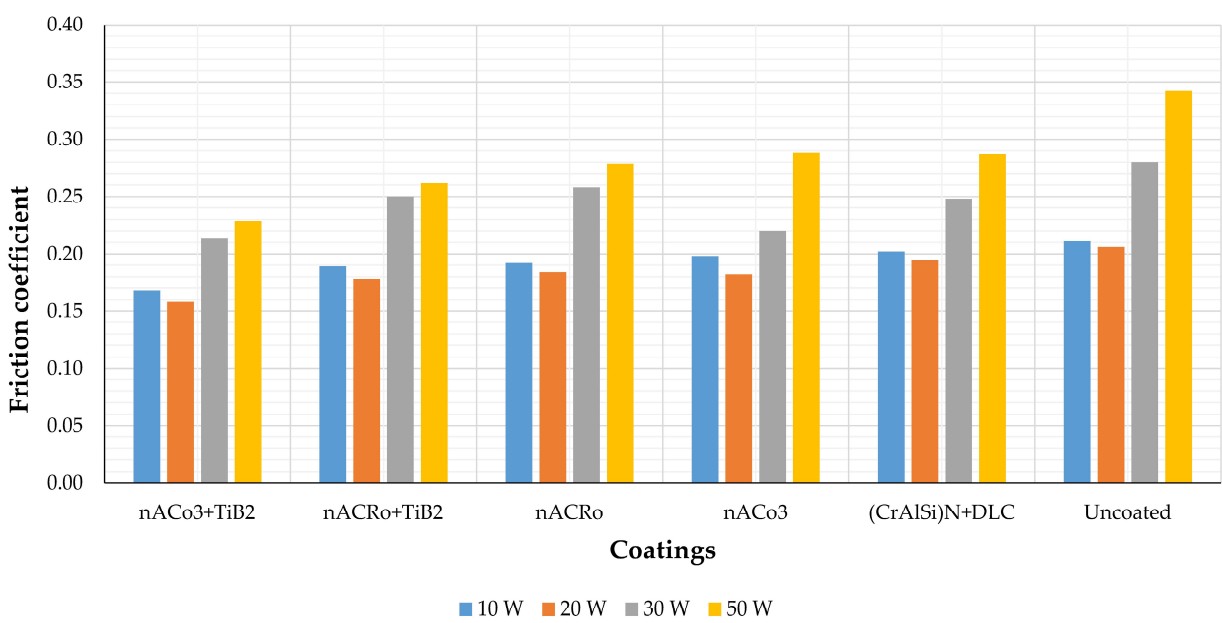

**Figure 13.** Influence of coating and laser processing power on friction coefficient of «EP99-H10F» pair with layers after laser processing at different strengths (at $p = 1$ MPa; $\theta = 700$ °C).

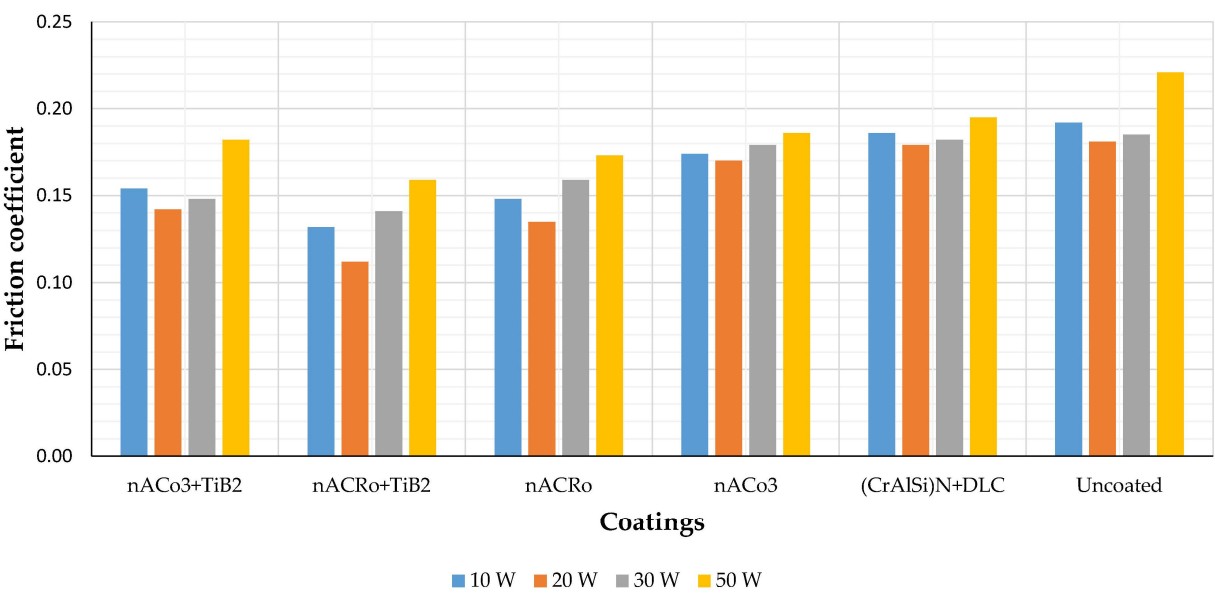

**Figure 14.** Effect of coating and laser processing power on the friction of «EK61-H10F» with layers after laser processing at different strengths (at $p = 1$ MPa; $\theta = 700$ °C).

The analysis of the obtained experimental data showed that the lowest coefficient of friction under these test conditions is provided by the following coatings for the pair «EP99—H10F», respectively: nACo3+TiB$_2$ (friction coefficient by 21% less compared to nACRo+TiB$_2$); nACRo+TiB$_2$ and nACRo, for the pair «EK61-H10F», respectively: nACRo+TiB$_2$ (friction coefficient by 25% less compared to nACRo); nACRo; nACo3+TiB$_2$. Based on the above, further experimental studies and laser processing of the tool material with different wear-resistant coatings were carried out at a power value equal to 20 W. Figures 15–20 show the results of adhesion studies—the dependence of the adhesion component of the coefficient of friction on the contact temperature for different coatings. Analysis of the obtained experimental data showed that for all the studied coatings after laser treatment with increasing temperature, the adhesion component of the friction coefficient monotonically increases and has a minimum value similar to the tests on the high-temperature

tribometer «Nanovea TRB», i.e., the best performance on tribotechnical parameters. i.e., the best indicators on tribotechnical characteristics correspond at temperature 650 °C: for pair «EP99-H10F»—nACo3+TiB$_2$ with a decrease on the average of 11% and for pair «EK61-H10F»—nACRo+TiB$_2$ with a decrease of the average on 8%.

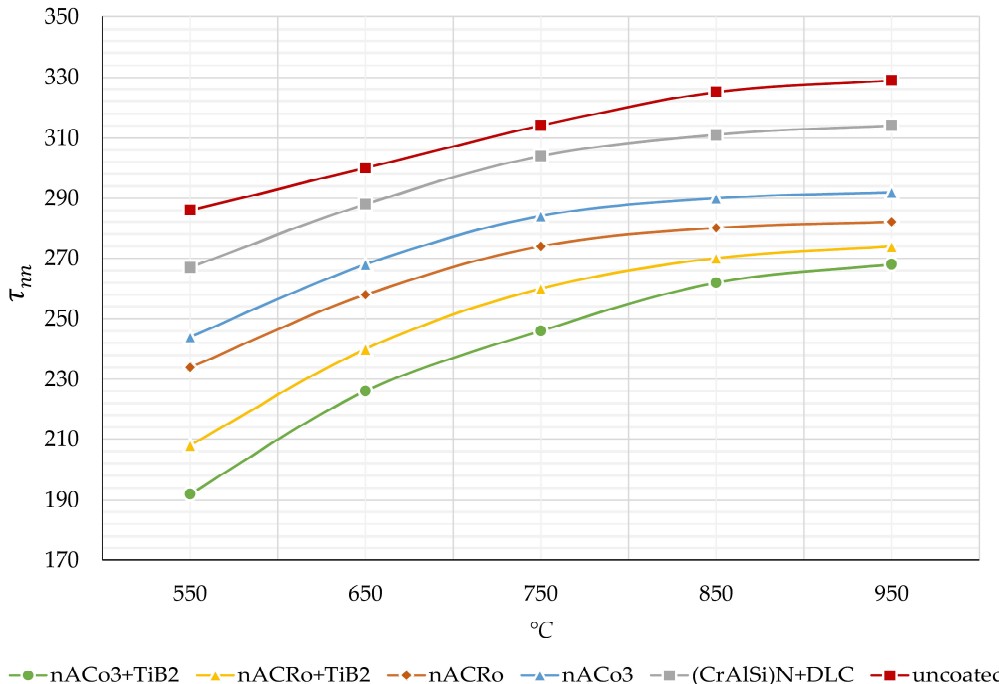

**Figure 15.** «EP99-H10F» Shear tangent stress of contact at adhesive interaction.

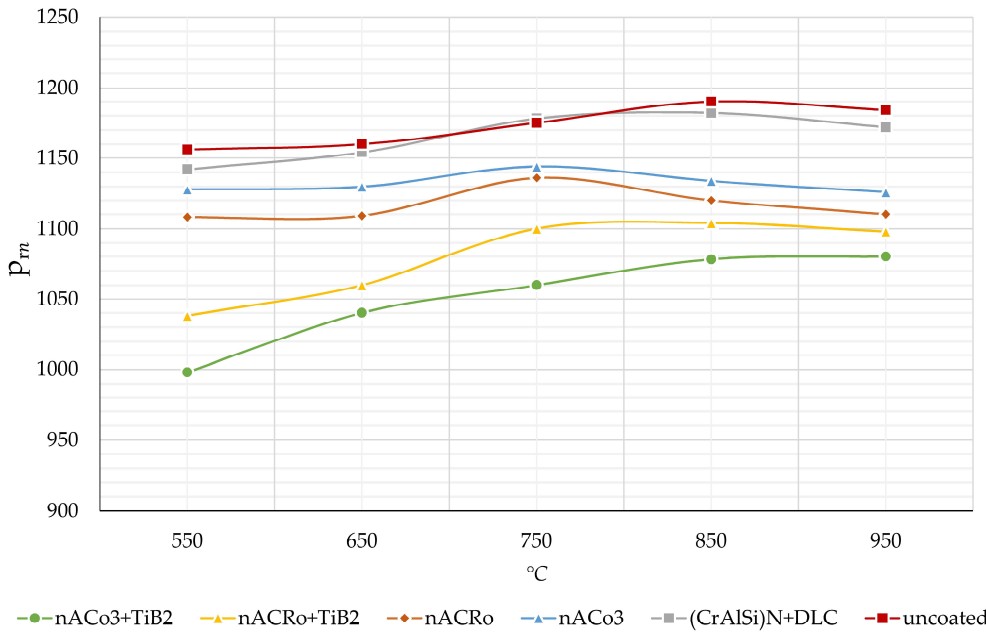

**Figure 16.** «EP99-H10F» Specific load per unit of contact area.

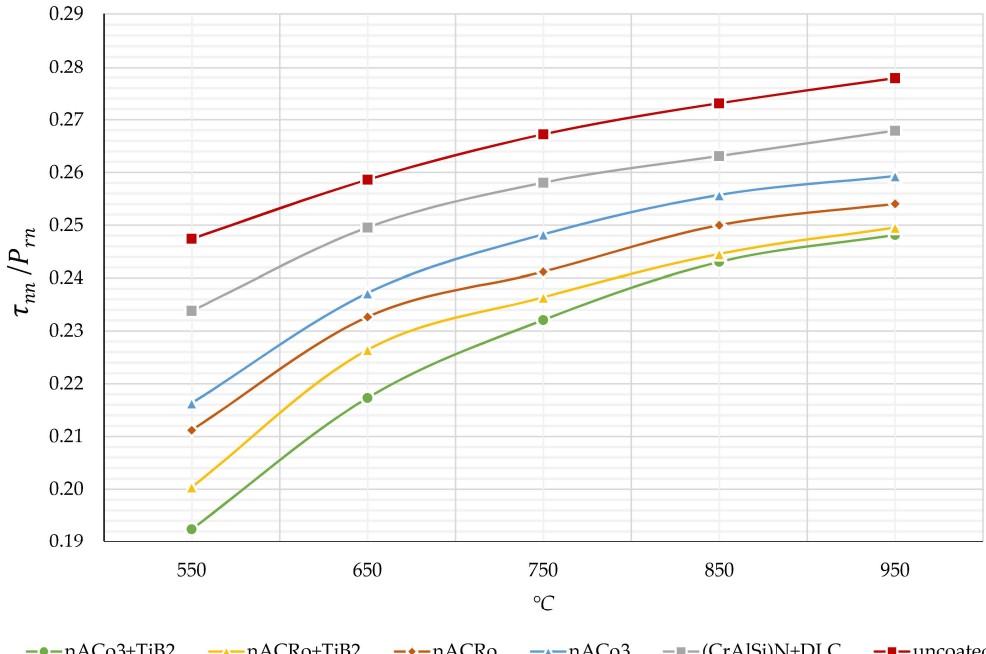

**Figure 17.** Temperature dependence of friction characteristics of plastic contact of «EP99-H10F» with coatings after laser treatment at 20 W power.

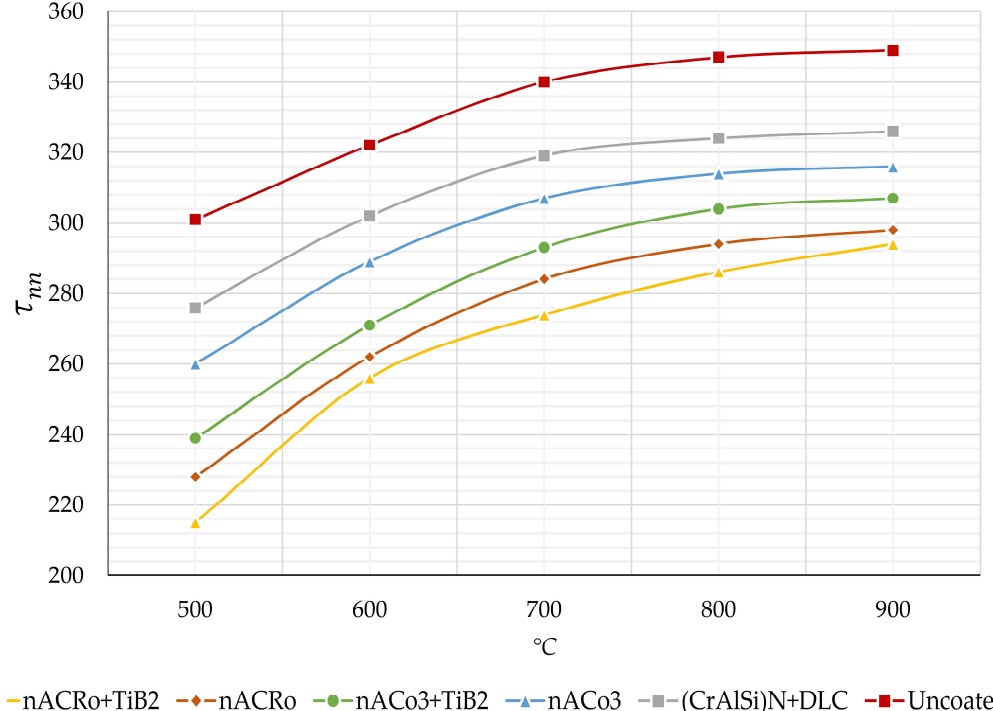

**Figure 18.** «EK61-H10F» Shear tangent stress of contact at adhesive interaction.

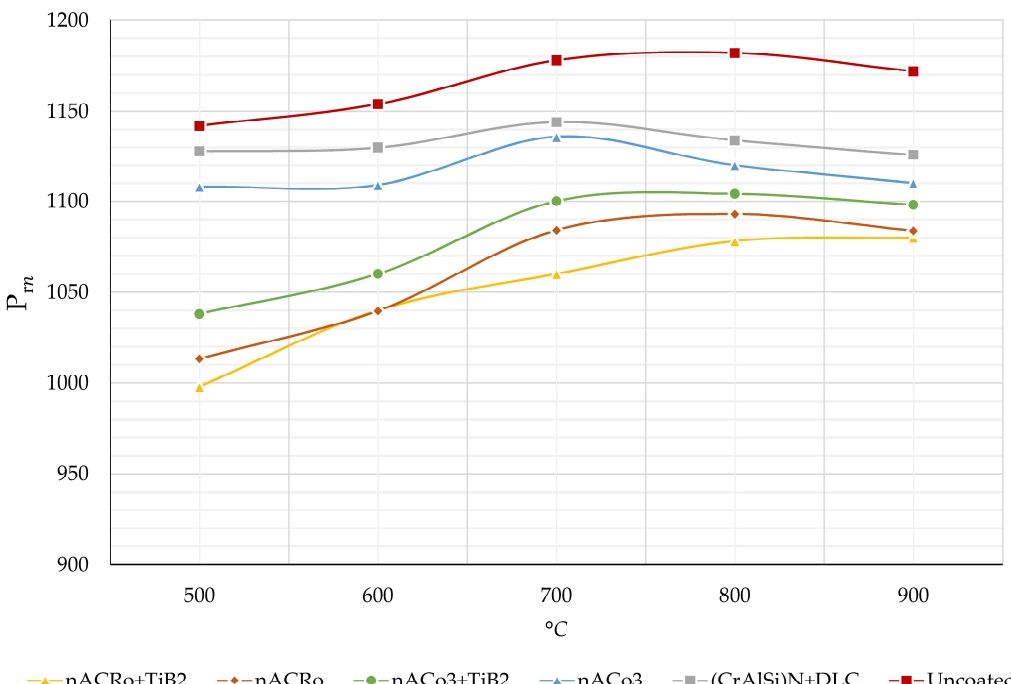

**Figure 19.** «EK61-H10F» Specific load per unit of contact area.

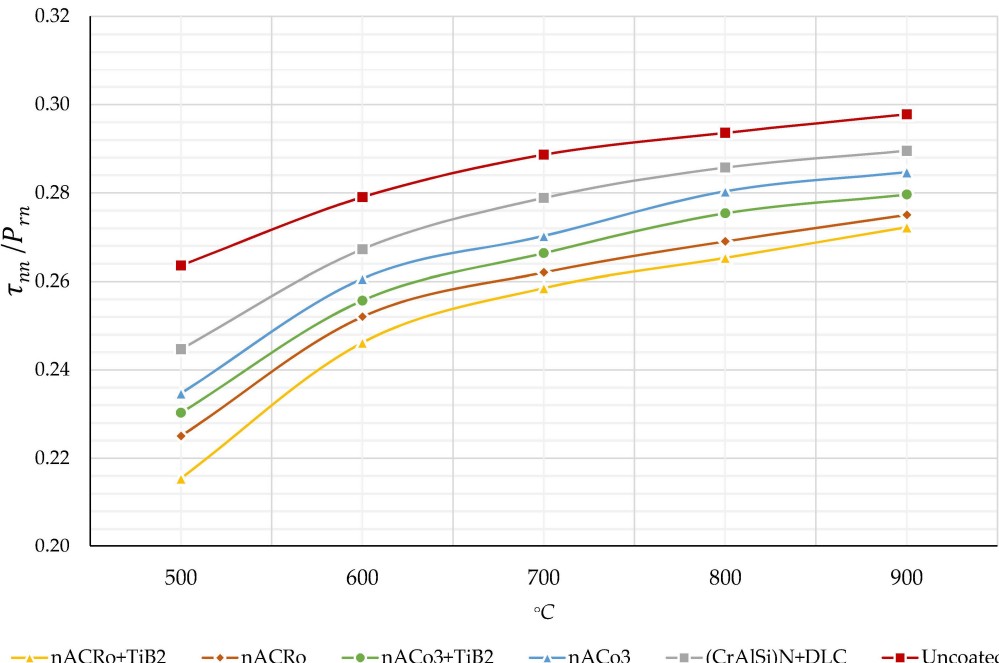

**Figure 20.** Temperature dependence of friction characteristics of plastic contact of «EK61-H10F» with coatings after laser treatment at 20 W power.

The results of temperature-force tests are presented in Figures 21–24. According to the effects of temperature-force tests, it is established that

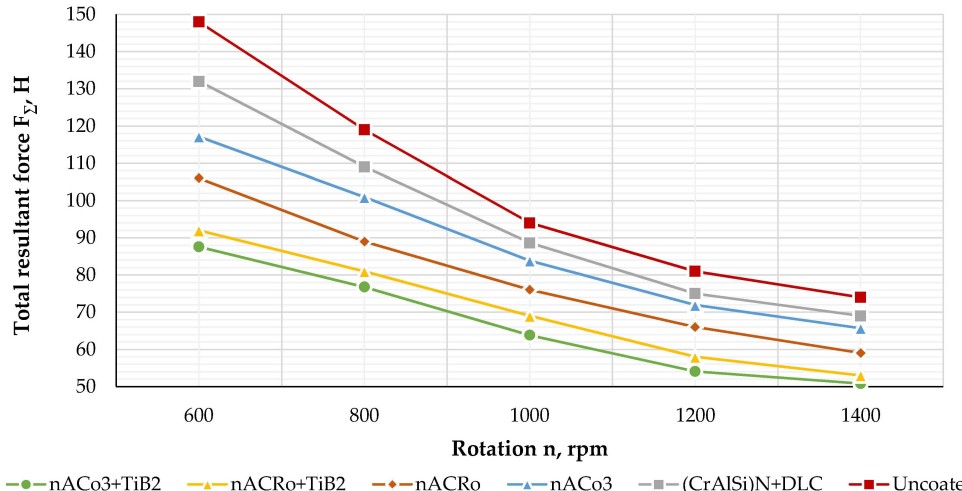

**Figure 21.** Effect of tool speed during EP99 milling on the tangential component of cutting force after laser machining different coatings.

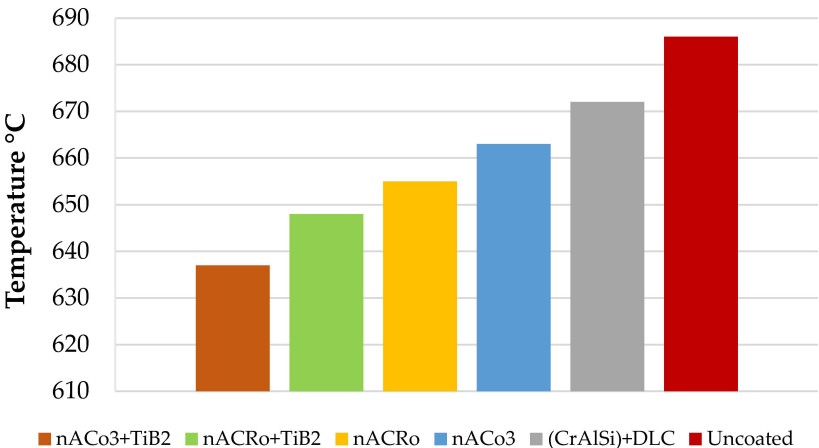

**Figure 22.** Dependence of cutting temperature on laser processing of different coatings at processing of EP99.

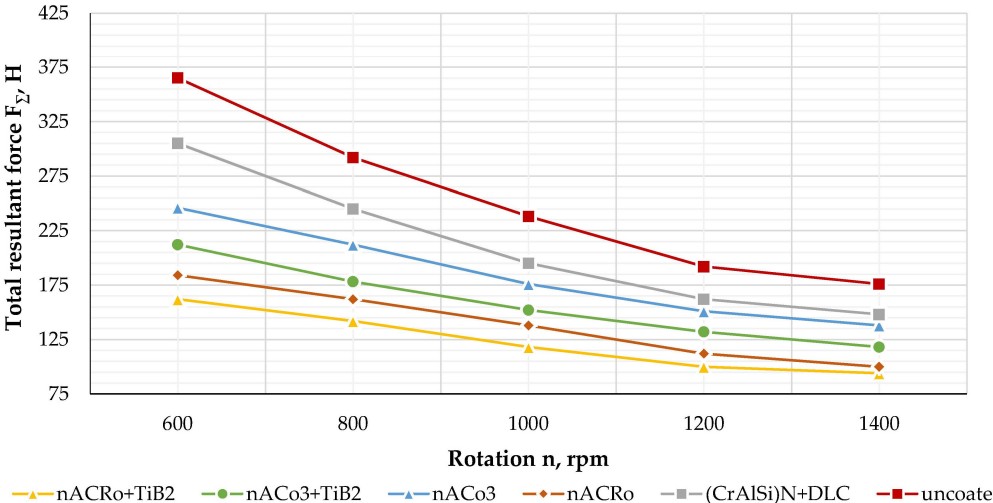

**Figure 23.** Effect of tool speed during EK61 milling on the tangential component of cutting force after laser machining different coatings.

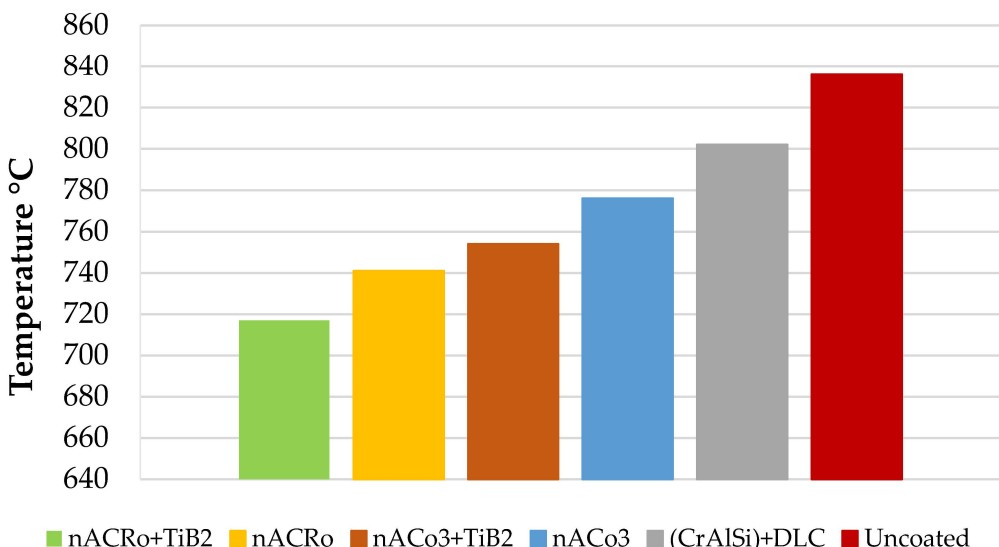

**Figure 24.** Dependence of cutting temperature on laser processing of different coatings at processing of EK61.

- with increasing tool rotation speed during milling, the tangential component of cutting force monotonically decreases both at the processing of EP99 and EK61;
- the lowest values are provided by the application of coatings: nACo3+TiB$_2$; nACRo+TiB$_2$ when milling EP99 and nACRo+TiB$_2$; nACRo when milling EK61;
- the lowest temperature stress in the contact zone is provided at milling EP99—nACo3+TiB$_2$ and EK61—nACRo+TiB$_2$.

The majority of experimental data on tribotechnical characteristics of nanostructured multilayer composite coatings on tools, in particular, on wear and durability of cutting tools, were obtained by direct observation and measurement of wear zones on a mill. At the same time, as preliminary experiments and analysis of literature data [30,31] have shown, in the conditions of semi-finishing cutting, the defining element of tool wear is the wear chamfer on its rear surface. The analysis of the back surface wear profile performed in the study [14,28,31] showed that the lowest variability of the wear measurement results is characterized by the average back surface wear along the leading cutting edge. This parameter at constant values of the front and back angles of the cutting wedge reflects the dimensional wear resistance of the tool. Based on the above, the average width of the rear surface wear chamfer (excluding notches) was used as the investigated tool wear parameter.

Wear resistance tests of cutters without and with coatings at different modes of laser processing were carried out, which are presented in Figures 25–28. The analysis of experimental data allowed us to determine the most favorable composition of coatings and the technology of their application. It has been established that:

- the most effective coatings in terms of contact processes on a single contact spot and adhesion interaction coefficient are the coatings applied on the «*Platit π411*» unit: nACo3; nACRo; nACo3+TiB$_2$; nACRo+TiB$_2$;
- the lowest coefficient of adhesion interaction between the tooling material and the processed material in the whole temperature range under study is provided by such coatings nACo3+TiB$_2$ for EP99 and nACRo+TiB$_2$ for EK61, which confirms the best adhesion of the coating with the substrate and allows using them for high-speed milling.
- the most extended length of the cutting path, respectively, and the period of durability provided at milling EP99—coating nACo3+TiB$_2$ by 28% in comparison with other coatings; at milling EK61—coating nACRo+TiB$_2$ by 13%.

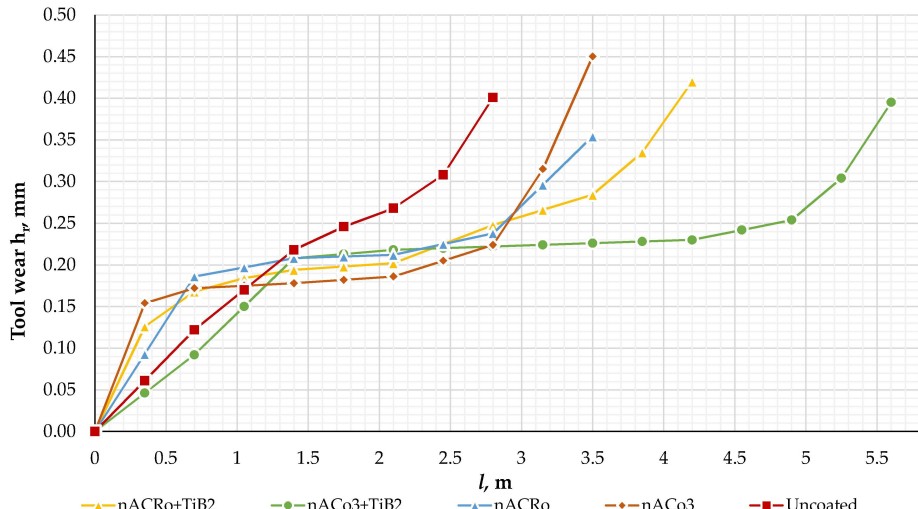

**Figure 25.** Effect of cutting path length on back surface wear during milling of «EP99-H10F» with different coatings after laser machining.

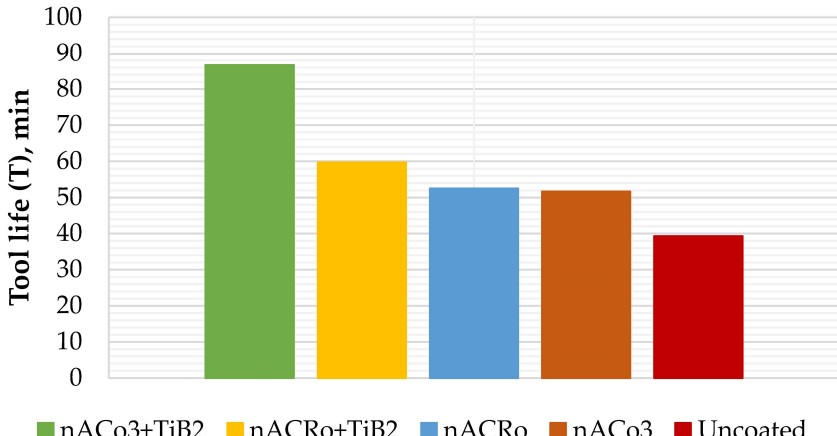

**Figure 26.** Durability period of metal-cutting tools with different coatings after laser treatment during milling of «EP99-H10F».

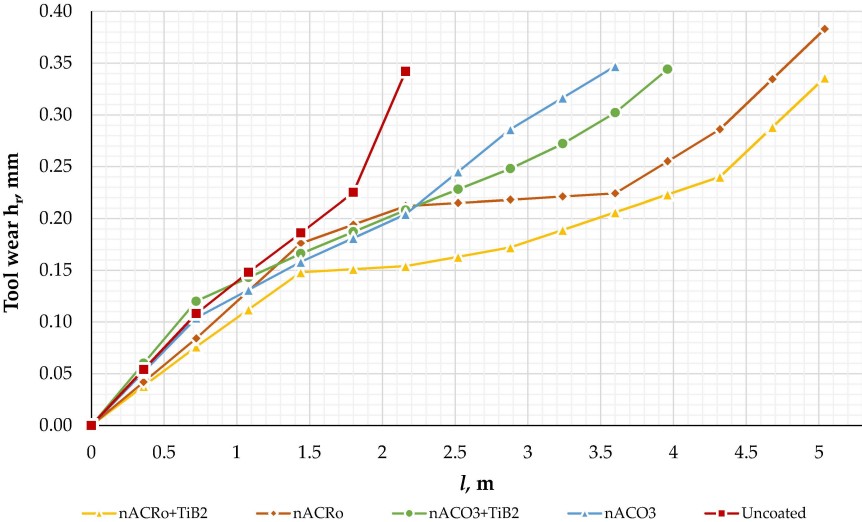

**Figure 27.** Effect of cutting path length on back surface wear during milling of «EK61-H10F» with different coatings after laser machining.

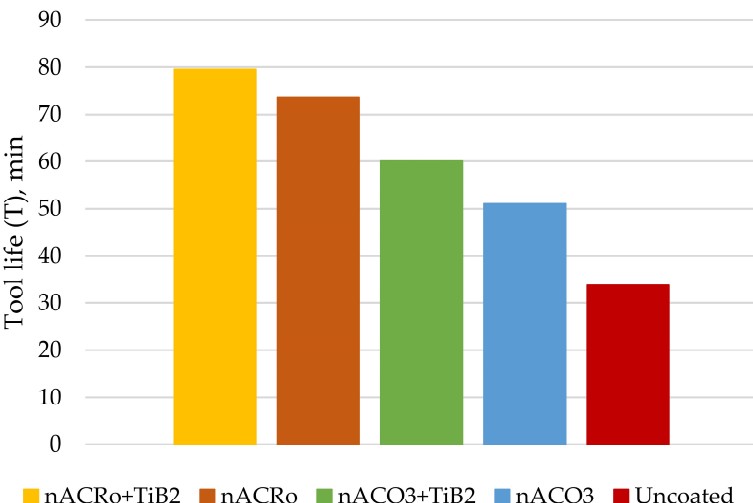

**Figure 28.** Durability period of metal-cutting tools with different coatings after laser treatment during milling of «EK61-H10F».

## 5. Conclusions

The paper analyzes modern approaches and methods of increasing the efficiency of blade machining of hard-to-machine chromium-nickel alloys widely used in aircraft parts and units. Innovative multilayer nanostructured coatings with subsequent laser processing of tool-cutting wedge surfaces with different textures have been developed and obtained, which allow efficient blade machining of hard-to-machine alloys at increased cutting modes. Also, methods of complex experimental research of operational properties of the received wear-resistant coatings of laser texturing are offered and approved, allowing to carry out including express estimation of them on tribometer and adhesion installation, which will reduce essentially duration and expenses for the testing process.

According to the results of experimental research of tribotechnical characteristics on the high-temperature tribometer «Nanovea TRB» the extreme character of dependence of friction coefficient on the laser computing power for both pairs: «EP99-H10F» and «EK61-H10F» with the minimum at the power of 20 W, as well as the decrease of friction coefficient value by 21% for the pair «EP99-H10F» with $nACo3+TiB_2$ coating and by 25% for the pair «EK61-H10F» with $nACRo+TiB_2$ coating was established.

Performed tests of tribotechnical parameters on the adhesion unit allowed us to establish that with increasing contact temperature, the adhesion component of the coefficient of friction monotonically increases and has a minimum value similar to tests on the high-temperature tribometer «Nanovea TRB», i.e., the best performance on tribotechnical parameters. At the same time, the best tribotechnical characteristics are provided at the contact temperature of 650 °C: for the pair «EP99-H10F»—$nACo3+TiB_2$ with an average decrease on the average on 11% and for pair «EK61-H10F»—$nACRo+TiB_2$ with a decrease of 8%.

The results of the research of contact processes: wear resistance of cutting tools, temperature, and components of cutting force during milling of heat-resistant and heat-hardness alloys EP99 and EK61 by carbide milling cutters H10F with wear-resistant coatings confirmed the effectiveness of the subsequent application of additional laser treatment of working edge surfaces and the possibility of increasing their by an average of 15%–20%. The results of temperature-force studies have shown that with increasing tool rotation speed during milling, the tangential component of the cutting force is the determining (main) component and monotonically decreases both during EP99 and EK61; the lowest values are provided during coating application: $nACo3+TiB_2$; $nACRo+TiB_2$ when milling EP99 and $nACRo+TiB_2$; nACRo when milling EK61; the lowest temperature stress in the contact zone is provided when milling EP99—$nACo3+TiB_2$ and EK61—$nACRo+TiB_2$, which is

explained by tribodegradation of titanium diboride at elevated cutting temperatures and critical values of the cutting force components.

In general, the increase of wear resistance of milling cutters is achieved by providing good adhesion between the coating and the tool substrate, reducing the coefficient of adhesion component of friction between the tool with wear-resistant coating and the material being machined, reducing and providing a favorable temperature and force regime, in particular, during cutting. Also, there is a high probability of the formation of amorphous-like secondary structures with the effect of hardening of a cutting wedge and lubrication of contact surfaces of a cutting tool with multilayer nanostructured wear-resistant coatings under certain temperature-force conditions in the contact zone «tool-workpiece, » in consequence of the manifestation of self-organization of friction and wear processes. In this case, the most effective are multilayer composite nanostructured wear-resistant coatings obtained on modern installations of the «Platit» series.

**Author Contributions:** Conceptualization: M.S.M.; methodology: A.V.G.; software: A.M.M.; validation: A.V.G.; formal analysis: A.V.G. and M.S.M.; investigation: A.P.M., A.S.G. and R.S.K.; resources: M.S.M.; data curation: A.S.G.; writing—original draft preparation: A.P.M.; writing—review and editing: M.S.M.; visualization: A.M.M.; supervision: A.V.G.; project administration: M.S.M.; funding acquisition: M.S.M. All authors have read and agreed to the published version of the manuscript.

**Funding:** This work is funded by the state assignment of the Ministry of Science and Higher Education of the Russian Federation, Project No. FSFS-2023-0003.

**Institutional Review Board Statement:** Not applicable.

**Informed Consent Statement:** Not applicable.

**Data Availability Statement:** Data sharing does not apply to this article.

**Acknowledgments:** The study was carried out on the equipment of the center of collective use of MSUT «STANKIN» supported by the Ministry of Higher Education of the Russian Federation (project No. 075-15-2021-695 from 26 July 2021, unique identifier RF 2296.61321X0013).

**Conflicts of Interest:** The authors declare no conflict of interest.

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
