# Peer review of "Effect of Surface Modification via Laser Irradiation on the Operability of Carbide End Mills When Cutting Aircraft Alloys"

_coatings, doi:10.3390/coatings13111823_

Round 1

Reviewer 1 Report

Well written manuscript. Suggestions for improving the manuscript are as follows

1. The abstract is generally written. Emphasize the applied methods, main results and conclusions in the abstract.

2. The scientific contribution and scientific hypothesis of this research should be explicitly written.

3. How are the parameters of laser processing selected? Why are they representative of this research?

4. Does additional laser treatment have any disadvantages?

5. Is it possible to estimate the measurement uncertainty of the results?

6. How can research be applied in practice? Are there any negative environmental impacts? What are the additional costs?

7. What are the limitations of the applied methodology in the research.

8. What are the directions of future research.

Author Response

We express our gratitude for your time, cooperation, and constructive comments on our work.

  1. The abstract is generally written. Emphasize the applied methods, main results and conclusions in the abstract.

Your remark is taken into account. In general, the essence of the work is to improve the efficiency of blade machining of hard-to-machine chromium-nickel alloys by milling with carbide milling cutters with developed innovative wear-resistant coatings followed by laser processing. At the preliminary stage to reduce labor-intensive and time-consuming wear-resistant tests, tribotechnical studies on Nanovea TRB tribometer and adhesion unit are carried out to determine the optimal modes of coating application and laser processing. After that a series of wear-resistance and temperature-force studies were carried out, confirming the increase of wear resistance of coated milling cutters after laser treatment due to the peculiarities of texture formation on the surface of the coated milling cutter and the formation of protective secondary structures in the form of amorphous-like hardened compounds and compounds with lubricating effect, according to the published joint works:

  • Kovaleva, I.O.; Grigoriev, S.N.; Gusarov, A.V. Non-disturbing boundary conditions for modeling of laser material processing. Physics Procedia, 2014, 8th international conference on laser assisted net shape engineering lane. 421–428. https://doi.org/10.1016/j.phpro.2014.08.145
  • Fox-Rabinovich, G.S.; Shuster, L.Sh.; Migranov, M.Sh. et al. Nano-crystalline filtered arc deposited (FAD) TiAlN PVD coatings for high-speed machining applications. Coat. Technol. 2004, 177-178, 800–811. https://doi.org/10.1016/j.surfcoat.2003.05.004
  • Fox-Rabinovich, G.S.; Shuster, L.Sh.; Migranov, M.Sh. et al. Elastic and plastic work of indentation as a characteristic of wear behavior for cutting tools with nitride PVD coatings. Thin solid films. 2004, 469-470, 505 – 512. https://doi.org/10.1016/j.tsf.2004.07.038
  • Stebulyanin, M.; Ostrikov, E.; Migranov, M.; Fedorov, S. Improving the Efficiency of Metalworking by the Cutting Tool Rake Surface Texturing and Using the Wear Predictive Evaluation Method on the Case of Turning an Iron–Nickel Alloy. Coatings 2022, 12, 1906. https://doi.org/10.3390/coatings12121906
  • Grigoryants, A.G.; Shiganov, I.N.; Perestoronin, A.V.; Misyurov, A.I.; Taksants, M.V.; Asyutin, R.D.; Usov, S.V. Kinetics of the WC particles motion in liquid steel during laser surface modification. Welding International 2021, 35:10-12, 408-414. https://doi.org/10.1080/09507116.2021.1901453
  • Shupenev, A.E.; Korshunov, I.S.; Krivosheev, A.V. et al. Laser treatment of heat exchanger plate surfaces for formation of DLC coatings. Metall. 2022, 654–659. https://doi.org/10.1134/S0036029522060246

  1. The scientific contribution and scientific hypothesis of this research should be explicitly written.

When conducting research, the main scientific hypotheses were:

- the possibility of increasing the efficiency of blade machining by cutting of hard-to-machine chromium-nickel alloys by modifying the surface of wear-resistant coatings on the cutting wedge by laser treatment, which would provide additional good adhesion of the coating with the substrate;

- formation both at laser processing and at cutting of such temperature and force conditions in the contact zone, contributing to the formation of secondary structures of two types: an amorphous-like one with hardening effect and another one with lubricating effect.

  1. How are the parameters of laser processing selected? Why are they representative of this research?

We apologize, the table is as follows:

Table 2. Laser treatment modes for indenter and cutting tool surfaces.

Power, W

10; 20; 30; 50

Spot diameter, µm

25; 40; 80

Pulse duration, ns

120; 125; 150; 200

Wavelength, nm

1064

Energy in the radiation pulse

up to 1 mJ

The parameters of laser processing were selected based on the analysis of literature data, as well as on the results of a large number of our own experimental studies on other types of both tool and processed material.

  • Ostrikov Е. А. Increase of efficiency of iron-nickel alloys machining on the basis of texturing of cutting tool working surface and method of predictive estimation of its wear. Abstract of dissertation. 2022. Р. 24.
  • Stebulyanin, M.; Ostrikov, E.; Migranov, M.; Fedorov, S. Improving the Efficiency of Metalworking by the Cutting Tool Rake Surface Texturing and Using the Wear Predictive Evaluation Method on the Case of Turning an Iron–Nickel Alloy. Coatings 2022, 12, 1906. https://doi.org/10.3390/coatings12121906
  • Grigoryants, A.G.; Stavertiy, A.Ya.; Bazaleeva, K.O.; Yudina, T.Yu.; Smirnova, N.A.; Tretyakov, R.S.; Misyurov, A.I. Laser surfacing of nickel-based composite war-resisting coatings reinforced with tungsten carbide. Welding International 2017, 31:1, 52-57, https://doi.org/10.1080/09507116.2016.1213039

  1. Does additional laser treatment have any disadvantages?

There were no special drawbacks and problems during laser processing, as well as experimental studies, as there was experience with the modes and the laser processing technology itself. There were noticed insignificant mixing of the coating with the tooling material, which we consider as a positive effect.

  1. Is it possible to estimate the measurement uncertainty of the results?

According to the recommendations for processing the results of experimental studies in carrying out wear-resistance, temperature-force tests of indenters and metal-cutting tools to ensure the reliability of the data obtained, up to 5 repeated experiments were performed.

  1. How can research be applied in practice? Are there any negative environmental impacts? What are the additional costs?

At present these researches are carried out and implemented in real concrete machine-building production, where there is a specialized tool shop with a complex of equipment for obtaining wear-resistant coatings (Platit 311, Platit 411 Plus), as well as Multifunctional SharpMark™ Fiber laser system. This laser system is of high environmental class and is housed in a specialized room with the necessary exhaust and safety equipment to ensure environmental safety. There are no additional costs.

  1. What are the limitations of the applied methodology in the research.

Limitations of the applied research methodology include: small tools less than 5 mm; tools with internally positioned cutting edges; ceramic tool material base, and ceramic coated tools. 

  1. What are the directions of future research.

As the main directions of future research we would like to mention the following:

 - To expand the scope of application of laser processing of wear-resistant coatings on turning tools, on drills, on taps, on reamers, on broaches, etc;

 - at different values of geometrical parameters of the cutting wedge of the tool;

 - as applied to tools and modern wear-resistant coatings for machining other hard-to-machine materials, in particular, titanium alloys, heat-resistant foundry alloys, high-strength and dispersion-hardening steels;

- metallographic studies of surfaces of indenters and milling cutters with various coatings using modern SEM and AES microscopy, secondary ion-mass microscopy, EELFS spectroscopy, for the formation of amorphous-like structures in the form of aluminum titanium oxides and others, both before wear-resistance and temperature-force tests, during and after the process, as well as at higher cutting regimes (there is a possibility that this will allow machining in higher regimes to minimize the intensity of wear of the cutting tool);

- modeling of temperature-force parameters for different types of wear-resistant coatings and laser surface treatment technologies for cutting tool surfaces for the purpose of prediction and diagnostics for other difficult-to-machine materials and blade machining methods.

Reviewer 2 Report

The manuscript centers on tailoring coatings through optimal processing parameters to augment the wear resistance of cutting tools. While it comprehensively covers experimental design, characterization, and performance evaluation, indicating a potential for publication, there are substantial areas needing improvement. Currently, the manuscript reads more like a routine scientific report rather than a compelling presentation of results for publication. The authors must accentuate their key findings, insights, and unique contributions more prominently.

1. The section titled "Theoretical and experimental prerequisites," currently nestled within the introduction, suffers from redundancy, making it challenging for readers to follow. It's essential for the authors to restructure this section, ensuring it's both succinct and informative to maintain reader engagement.

2. In line 274, kindly remove the repeated term "heat-resistant" to enhance clarity and avoid redundancy.

3. Please specify the units for the chemical composition in Table 1.

4.  The "Materials, equipment and technology of experimental studies" section needs substantial reorganization for clarity. I suggest the authors use more concise language and consider relocating some of the detailed content to supplemental documents to simplify the main text.

5. Are the visuals in Figure 10 derived from SEM imaging? If they are, it would be beneficial to specify which SEM was employed for the characterization. Additionally, the data area in these images is quite limited, making it difficult to discern the scale bar and any accompanying measurements. I noticed a similar issue in Figure 12; please address this as well.

The manuscript's language necessitates substantial refinement. Presently, several sections come across as redundant, obfuscating the intended message and making comprehension difficult. It's imperative for the authors to streamline their writing for clarity and coherence.

Author Response

We express our gratitude for your time, cooperation, and constructive comments on our work.

  1. The section titled "Theoretical and experimental prerequisites," currently nestled within the introduction, suffers from redundancy, making it challenging for readers to follow. It's essential for the authors to restructure this section, ensuring it's both succinct and informative to maintain reader engagement.

Worked to reduce and restructure the main sections of our paper.

  1. In line 274, kindly remove the repeated term "heat-resistant" to enhance clarity and avoid redundancy.

We assumed different terms, unfortunately an unfortunate translation was made. It was about heat-resistant and heat-hardness materials. «heat-resistant and heat-hardness materials».

  1. Please specify the units for the chemical composition in Table 1.

The comment on Table 1 has been removed (percentage content).

  1. The "Materials, equipment and technology of experimental studies" section needs substantial reorganization for clarity. I suggest the authors use more concise language and consider relocating some of the detailed content to supplemental documents to simplify the main text.

Your comments have been taken into account; indeed, some points are overloaded with minor information.

  1. Are the visuals in Figure 10 derived from SEM imaging? If they are, it would be beneficial to specify which SEM was employed for the characterization. Additionally, the data area in these images is quite limited, making it difficult to discern the scale bar and any accompanying measurements. I noticed a similar issue in Figure 12; please address this as well.

Yes, the images are derived from the SEM images in Figure 11 for the Nanovea tribometer indenters and adhesion unit, and for the carbide milling cutters with wear resistant coatings in Figure 13. The Tescan Vega 3.0 LMH SEM rig in Figure 10 was used for the study.

Round 2

Reviewer 2 Report

I believe the authors have addressed the majority of my concerns, and the manuscript has seen significant improvement. However, there are still a few areas that require further modifications by the authors before I can recommend this manuscript for publication:

1. In Table 1, kindly clarify whether the values are presented in weight percent or atomic percent.

2. There's no need for the authors to include the SEM instrument image in Figure 4. Instead, please provide the instrument details within the text.

3. On line 173, ensure the correct subscript notation is used, for example, TiB₂.

Author Response

We express our immense gratitude, constructive comments on our work and cooperation.

  1. In Table 1, kindly clarify whether the values are presented in weight percent or atomic percent.

In weight percentages, (wt%). We corrected.

  1. There's no need for the authors to include the SEM instrument image in Figure 4. Instead, please provide the instrument details within the text.

We removed the image of the device and gave the main data as text.

  1. On line 173, ensure the correct subscript notation is used, for example, TiB.

We corrected the signature TiB2 to TiB2 , nACo3 coating was not corrected – it is a designation accepted in a publicly available form.
